# Zn, Cu, and Fe Concentrations in Dehydrated Herbs (Thyme, Rosemary, Cloves, Oregano, and Basil) and the Correlation with the Microbial Counts of *Listeria monocytogenes* and Other Foodborne Pathogens

**DOI:** 10.3390/foods9111658

**Published:** 2020-11-12

**Authors:** José María García-Galdeano, Marina Villalón-Mir, José Medina-Martínez, Lydia María Vázquez-Foronda, Jessandra Gabriela Zamora-Bustillos, Ahmad Agil, Sofía María Fonseca Moor-Davie, Miguel Navarro-Alarcón

**Affiliations:** 1Department of Nutrition and Bromatology, School of Pharmacy, University of Granada, 18071 Granada, Spain; info@farmaciagaldeano.com (J.M.G.-G.); marinavi@ugr.es (M.V.-M.); pepemm97@correo.ugr.es (J.M.-M.); lydiavazfor8@correo.ugr.es (L.M.V.-F.); j.gaby_@hotmail.com (J.G.Z.-B.); sofiafonseca@correo.ugr.es (S.M.F.M.-D.); 2Nutrition and Food Technology Institute of Granada, University of Granada, 18010 Granada, Spain; 3Department of Pharmacology, Neurosciences Institute, School of Medicine, University of Granada, 18010 Granada, Spain; aagil@ugr.es

**Keywords:** Zn, Cu, and Fe concentrations, dehydrated herbs, microbial counts for foodborne pathogens

## Abstract

Zn, Cu, and Fe concentrations were measured in dehydrated herbs (thyme, rosemary, cloves, oregano, and basil) marketed in bulk or packaged in glass or polyethylene terephthalate (PET). Microbial counts of *Listeria monocytogenes* and other five foodborne pathogens were also checked when herbs were previously added to the growing media. The highest mean concentrations were found in basil for Zn and Cu, and in thyme and basil for Fe; the lowest ones for these minerals were in cloves (*p* < 0.05). Basil had significantly higher microbial counts in five of the six foodborne pathogens studied (*p* < 0.05). Cloves have the best hygienic quality as there is no microbial growth of *L. monocytogenes*, *Clostridium perfringens*, and *Bacillus cereus*; they therefore could be used as a natural preservative in food. Aromatic herbs marketed in bulk showed a significantly higher microbial count (*p* < 0.05). Zn, Cu, and Fe concentrations were positively correlated with microbial growth for *L. monocytogenes*, *C. perfringens*, *B. cereus*, and psychrophilic microorganisms (*p* < 0.05), so they could act as a growing factor for the foodborne pathogens.

## 1. Introduction

Aromatic herbs are minority components, as far as consumption is concerned, in the daily diet [1] (1.5 g/day in European countries [2]). Their traditional use is associated with the seasoning of culinary preparations to facilitate desirable flavors, aromas, and/or colors, being the main determinants of the sensory quality of food [1,2]. Another extended use of these herbs, increasing the amount added to food, is the masking of undesirable flavors when they have been submitted to an inadequate state of preservation. Aromatic herbs have also been used, for their content of essential oils and polyphenolic antioxidant compounds, in traditional medicine [1,3,4,5], and as natural food preservatives due to the antimicrobial and antifungal action of their constituent essential oils [6,7,8,9,10,11]. In recent years, ethanolic extracts of rosemary at 1% in active packaging have been proposed as components of a protective film of biodegradable serum protein, as they present antimicrobial activity against *L. monocytogenes* [12]. Despite this, the low solubility, bioavailability, and high volatility of essential oils are important limitations on their application in food as antibacterial agents [8].

In recent years, numerous studies have linked the bioactive compounds present in these aromatic herbs, especially in their essential oils, to a preventive effect against various pathologies related to oxidative stress, indicating their anti-inflammatory, antidiabetic, and antihypertensive effect [13]. In this sense, Papageorgiou et al. [14] refer to an increase in antioxidant capacity and therefore a delay in lipid oxidation in turkey tissues after the incorporation of oregano oil in the diet (200 mg/kg of diet). Regarding basil essential oils, other researchers indicate that these have antioxidant and antifungal activities, and that they are maximal in essential oils obtained at concentrations of 0.04 mg/L of Cu fertilizer [15] and 0.095 mg/L of Zn fertilizer [16]. Other researchers refer to oregano essential oils as an alternative to butylated hydroxytoluene to extend the shelf life of packaged meat products like ground beef [17].

The antioxidant effect of rosemary essential oils against Fe-induced oxidation has been observed in homogenized pig longissimus dorsi muscle with a dose-dependent response [18]. Therefore, aromatic herbs, as a source of phenolic antioxidants that minimize oxidative rancidity and increase the shelf life of foods, can meet the food industry’s need for natural and clean-labeled products [19]. In recent years, the use of synthesized nano-Zn oxide particles using clove extracts as antimicrobial agents has been studied [11].

Numerous studies have been carried out on aromatic herbs in relation to their heavy metal content, due to their ability to concentrate heavy metals from contaminated farmland; they may, therefore, may be a source of toxic trace metals in the human body [1]. The accumulation of elements in plants until they reach toxic levels depends on the varieties of plants, and on the environmental and climatic conditions under which they are cultivated [20]. It is necessary to know the geographical origin of food (in particular of herbs) in order to guarantee the health of the final consumer [21]. The metals analyzed in this work, Zn, Cu, and Fe, are essential elements related to important functions in biological systems and in the human body, as long as their intake does not exceed the upper limits set by the Institute of Medicine [22], in which case they could be associated with toxic effects [23]. However, given the low daily dietary intake of aromatic herbs (0.8‒1.5 g/day), it is difficult for these to contribute to toxic levels, as indicated for Fe determined in aromatic plant spices growing in soils from an abandoned mine in Roalgar, SW Portugal [24]. Zn, Cu, and Fe are cofactors of the superoxide dismutase and catalase antioxidant enzymes, respectively. However, high levels of free Cu and Fe in the body act as pro-oxidants [25], resulting in pathologies like cardiovascular diseases, hypertension, type 2 diabetes mellitus, multiple sclerosis, etc. [24].

In food technology, during the processing of raw foods such as certain meat derivatives, it is common to use aromatic herbs, both fresh and dehydrated, which could be related, despite the preservative effect traditionally associated with their use [7,26], to foodborne pathogenic microorganisms, which can be responsible for episodes of food poisoning in final consumers. Several studies have shown that these herbs are sometimes exposed to a wide range of microbiological contaminants during and after harvest [27,28,29,30], and especially at points of sale and distribution, which reduces their hygienic quality and can make them a vehicle for foodborne diseases, especially when the food containing them is consumed raw or in culinary preparations that do not require lethal heat treatment [30]. Ferriccioni et al. [5] have indicated that dried thyme can be decontaminated before marketing and use by ultraviolet radiation treatment, which reduces the growth of aerobic mesophilic microorganisms, molds, and yeasts by more than 90%. Dogu-Baykut and Gunes [31] indicate that with this treatment, in addition to the microorganisms mentioned above, the development of *B. cereus* is also considerably reduced.

It is known that microorganisms in general, and foodborne pathogens in particular, require for their optimal development an adequate contribution of essential elements, and that Fe specifically is one of the stimulators of microbial growth, as has been indicated for *L. monocytogenes* [32] and in general for the development of any pathogenic microorganism [25], by acting as a growth factor. However, it has been indicated that microbiologically synthesized Ag‒Cu nanoalloys have antibacterial effects against, among others, *L. monocytogenes* at concentrations of 0.01 g/L [33], or, as Cu nanoparticles, increase the antifungal activity of thyme essential oils [10].

For all these reasons, in the present study, we intend to determine the levels of Zn, Cu, and Fe in five of the most common dehydrated aromatic herbs used in the Mediterranean diet, both for the preparation of salty and sweet dishes and whose consumption is widespread among the population of the Mediterranean arc (thyme (*Thymus vulgaris*), rosemary (*Salvia rosmarinus*), cloves (*Sycygium aromaticum*), oregano (*Origanum vulgare*), and basil (*Ocimum basilicum*)). At the same time, we will analyze the hygienic quality of these aromatic herbs by determining the microbial growth of six groups of the most common foodborne pathogens (*L. monocytogenes*, *Clostridium perfringens*, *B. cereus*, psychrophilic microorganisms, mesophilic microorganisms, and molds and yeasts), to know if the levels of the essential elements studied in the indicated aromatic herbs are related to the growth of these microorganisms. To this end, the marketing system used at the consumer and catering level (bulk sales, glass containers, and polyethylene terephthalate (PET) containers) will be taken into account. Aromatic herbs have traditionally been associated with an antimicrobial effect, which explains their inclusion in essential oils. However, comparison of the preservative capacity of the herbs studied in this manuscript when added to food not submitted to thermal final treatment has not often been performed. Additionally, it has not been verified whether some of them act directly as a vehicle for foodborne pathogens. The aim is to obtain data that will allow us to advise the food industry, and even catering establishments and consumers, about which aromatic herbs are the most advisable for use as food preservatives and condiments, and how the content of Zn, Cu, and Fe can influence this.

## 2. Materials and Methods

### 2.1. Sampling of Aromatic Herbs

The samples analyzed were dehydrated thyme, rosemary, cloves, oregano, and basil (*n* = 75) commercialized in bulk (*n* = 25), and packaged in glass jars (*n* = 25) or PET jars (*n* = 25) that were collected from different commercial establishments in Spain, aseptically in sterile bags, before microbiological analysis on the same day. Seventy-eight percent of the samples collected were of European origin; the remaining 22% were of unknown origin but sold in Spanish establishments.

Bulk samples were collected from herbalists, where the dried herbs are displayed to the public in baskets or boxes without any packaging system and sometimes in stalls by the highway. This entails a great deal of handling and a possible danger of contamination, both environmentally and by consumers themselves, who come to the baskets to smell them and sometimes even touch them with their hands.

The samples packed in glass and PET were collected from supermarkets, packaged, and hermetically sealed in order to avoid manipulation by users and consumers who come to these food centers. We have chosen glass and PET containers since they are materials with high barrier properties against gases and vapors in general. With this, we intend to see the difference in terms of the microbial quality of the food between herbs without any type of container that are sold in bulk, and packaged herbs that have been previously treated by the food industry, to ensure their hygienic quality. The container is a guarantee of that quality, as it acts as a barrier that prevents the passage of microorganisms from the environment, as well as certain gases (such as water vapor) that can favor microbial development.

### 2.2. Elemental Analysis in Samples of Aromatic Herbs

Three hundred milligrams of aromatic herb sample were mineralized by attack with HNO_3_ (66%) and HClO_4_ (60%) of suprapure quality (Merck, Darmstadt, Germany) in a thermostatized multisite mineralization block (Selecta, Barcelona, Spain). Mineralization was carried out in two stages: in the first stage, 4 mL of HNO_3_ were added by heating to 60 °C for 60 min, 90 °C for 60 min, and 120 °C for an additional 120 min; in the second stage, 3 mL of a mixture of HNO_3_‒HClO_4_ (4:1) were added by heating to 90 °C for 60 min, 120 °C for 60 min, and 130 °C for 90 min. The digest finally obtained was diluted to 10 mL with reagent-grade water (Milli-Q water obtained using the R015 Milli-Q system, Waters, Medford, MA, USA) in order to obtain the solution for analysis. The determination of total concentrations of Zn, Cu, and Fe in aromatic herb samples was performed by an atomic absorption spectrophotometer (AAS) instrument (Varian SpectraA, 140, Mulgrave, Australia). Calibration curves were previously prepared by diluting stock solutions of 1000 mg/L in 1% HNO_3_ for the analyzed elements (Merck, Darmstadt, Germany).

The accuracy and precision of Zn, Cu, and Fe measurement procedures (*n* = 10) were verified by testing the certified reference standard apple leaf powder from the National Institute for Standards and Technology (NIST) 1515 (Gaithersburg, MD, USA). No significant differences were found between the mean element concentrations determined in these materials and the certified concentrations (12.5 ± 0.68 and 12.7 ± 0.85, 5.19 ± 0.43 and 5.48 ± 0.35, and 84.7 ± 2.20 and 83.1 ± 3.00 µg/g for Zn, Cu, and Fe, respectively). Additionally, the accuracy of the methods was tested on the basis of recovery experiments, after complete digestion of spiked aromatic herb samples with different amounts of elements from the standard solutions [34]. The calculated recoveries for each element were between 96% and 101.3% in all cases. The limits of detection (LOD) of the method for the elements analyzed (3.2, 0.80, and 0.28 ng/mL, for Zn, Cu, and Fe, respectively) were calculated as previously reported [34]. The concentration (µg/mL) in samples was obtained by linear calibration. Every element was analyzed in triplicate in each of the samples of aromatic herbs.

### 2.3. Microbiological Analysis Methods

Microbial counts have been checked for *L. monocytogenes*, *C. perfringens*, and *B. cereus*, as microorganisms indicating a lack of hygienic quality in herbs. These bacteria can appear in food due to cross-contamination and a lack of hygiene during storage and handling. Mesophilic and psychrophilic microorganisms have been studied because it is common to find them in aromatic plant cultures. Molds and yeasts have also been studied; these microscopic organisms feed and perform their vital functions based on the decomposition of organic matter. The main problem with them is the production of mycotoxins.

For the microbial count, 1.5 g of each of the aromatic plants with which we are going to work were diluted in 9 mL of sterile buffered peptone water solution and, from these, decimal dilutions were made in the same way. These amounts of herbs were taken as a reference based on feasible portion sizes when planning a menu. The peptone water served as a dilution and pre-enrichment medium for most of our microorganisms (mesophilic, psychrophilic, *Bacillus*, and molds and yeasts). From the decimal dilutions, appropriate dilutions (1 mL) were sown in triplicate, using direct coating methods.

Counts of aerobic mesophilic bacteria were made by the pour-in-plate method, on plate count agar (PCA medium), followed by incubation at 30 ± 1 °C for 72 h [35]. Yeast and mold enumeration was performed by the pour-in-plate method using Sabouraud agar with chloramphenicol (Sabouraud dextrose agar, according to the standards learned for food and beverages in the National Center of Microbiology, Carlos III Institute, Madrid, Spain), followed by incubation at 25 ± 1 °C for 3–5 days [36]. For psychrophilic microorganisms, inoculation in plates with King agar A medium was used, followed by incubation for 24‒48 h at room temperature (20 °C).

In order to detect *L. monocytogenes*, according to ISO 11290:2017 [37], the studied samples (1.5 g) were weighed in Stomacher bags, diluted, and homogenized with 225 mL of Fraser (primary enrichment in half-Fraser broth: incubation for 25 ± 1 h); secondary enrichment in Fraser broth: incubation for 24 ± 2 h. After this time, sowing was carried out in *Listeria* plates of PALCAM *Listeria* agar 37 ± 1 °C for 24 ± 2 h (plus another 24 ± 2 h at 37 ± 1 °C).

For *B. cereus* count, ISO 7932:2005 standard [38] was followed by counting colonies in mannitol egg yolk polymyxin (MYP) agar medium. After sowing, the plates were incubated at 30 °C for 18–24 h. The count was carried out on plates containing fewer than 150 typical colonies (large, pink, with a precipitation halo). For confirmation, at least five presumptive colonies were taken and sown in plates on ram blood agar. After incubation at 30 °C for 24 h ± 2 h, *B. cereus* produces a positive hemolysis reaction.

Finally, *C. perfringens* was analyzed according to ISO 7937:2005 Standard [39], using ISO salt pepton water as the diluent. After inoculation of the sample in SC agar plates, it was set up for incubation in anaerobic conditions at 37 °C for 20 h ± 2 h. On plates with fewer than 150 colonies, five of these colonies were selected for confirmation by seeding in degassed lactose gelatin agar and incubating in anaerobic conditions at 37 °C for 24 h.

### 2.4. Statistical Analysis

The homogeneity of variance was assessed using Levene’s test and the normal distribution of data with the Kolmogorov‒Smirnov test. The ANOVA test was used to analyze parametric data and the Kruskal‒Wallis test to analyze nonparametric data. Linear correlations between elements and microbial growth measured in aromatic herbs were evaluated by computing the correlation coefficient by Pearson (for normally distributed data) or Spearman linear correlation (for non-normally distributed data). The significance level was set to 5% (*p* < 0.05) in all tests. SPSS 22.0 for Windows (IBM SPSS Inc., Armonk, NY, USA) was used for the data analyses.

## 3. Results

The content of Zn, Cu, and Fe varies depending on the aromatic herb (Table 1, Figure 1A,B). Specifically, the mean Zn and Cu concentrations were significantly different among all aromatic herbs (*p* < 0.001), with the exception of thyme and oregano. Specifically, the highest mean level was found in basil, which was significantly higher than that of thyme and oregano; these, in turn, were significantly higher than the level measured in rosemary, and the latter was higher than that determined in cloves (*p* < 0.001). In general, the highest average concentrations of Zn and Cu were found in basil and the lowest in cloves (Figure 1A). However, the commercialization system (glass or PET packaging, or bulk sale) does not influence the content of Zn and Cu in the analyzed aromatic herbs considered together (*p* > 0.05; Table 2).

In relation to the average levels of Fe (Table 1, Figure 1B), although there are no statistically significant differences between those in thyme and basil (*p* > 0.05), they are significantly higher (*p* < 0.001) than those of the other aromatic herbs. In turn, the average concentration of Fe measured in oregano was significantly higher than that of rosemary, and that of the latter and oregano, in turn, with respect to that of cloves (*p* < 0.001). The marketing system (glass or PET packaging, or bulk sale) significantly influenced the Fe content of the analyzed aromatic herbs considered together (*p* < 0.05; Table 2). Specifically, the levels determined in the commercially available herbs packaged in PET were significantly lower (*p* < 0.01).

Figure 2A,B, Figure 3A,B and Figure 4A,B show the values of microbial growth for the six groups of microorganisms considered according to the different aromatic herbs studied. It can be seen globally that basil is the aromatic herb that presents superior growth levels for the majority of the microorganisms.

In the case of *L. monocytogenes* (Figure 2A), the average growth value in thyme was significantly higher than that determined for the other aromatic herbs, followed by basil, which was also significantly higher (*p* < 0.001) than the average growth level obtained with rosemary and oregano (since no statistically significant differences were found between rosemary and oregano, *p* > 0.05). Finally, in cloves, this microorganism presented zero growth, significantly lower than the remaining aromatic herbs (*p* < 0.001).

In relation to *C. perfringens*, its average growth (Figure 2B) in basil was significantly higher than in the rest of the aromatic herbs (*p* < 0.01), and that of cloves was null and significantly lower than the other aromatic herbs (*p* < 0.01). In addition, the average growth in rosemary was significantly lower than that of thyme and oregano (*p* < 0.01).

The average growth value of *B. cereus* in basil was also significantly higher than that determined for rosemary, thyme, and cloves (*p* < 0.001; Figure 3A). In addition, thyme and rosemary (with no differences between the mean growth levels of this microorganism) showed significantly higher growth (*p* < 0.001) than cloves (where cloves are null). On the other hand, the mean growth level of *B. cereus* in oregano was significantly higher than in rosemary and thyme (*p* < 0.001).

With respect to the growth of psychrophiles, basil is the aromatic herb that most facilitates their development in comparison to thyme, rosemary, oregano, and cloves (*p* < 0.001; Figure 3B). In addition, the growth of psychrophiles is significantly greater in the presence of thyme than cloves (*p* < 0.05).

For mesophilic microorganisms, basil also had the highest growth rate of all the aromatic herbs (*p* < 0.001). However, oregano had significantly less growth of mesophilic microorganisms compared to the other aromatic herbs, with the exception of cloves (Figure 4A).

Finally, in relation to molds and yeasts (Figure 4B), the addition of basil to the culture medium leads to a statistically significantly higher (*p* < 0.001) growth rate of these microorganisms in comparison to all other aromatic herbs; the only exception is rosemary, for which there are no significant differences (*p* > 0.05).

However, we appreciate that the microbial growth values were influenced by the commercialization system of the analyzed aromatic herbs (Table 3). Specifically, it was observed that aromatic herbs sold in bulk were associated with significantly higher microbial growth than those packaged in PET and glass (*p* < 0.05). Moreover, aromatic herbs sold in bulk were associated with significantly higher microbial growth than glass-packaged herbs for *L. monocytogenes*, *B. cereus*, and mesophilic microorganisms (*p* < 0.05). Finally, aromatic herbs sold in glass containers were associated with significantly higher microbial growth than those packaged in PET for molds and yeasts (*p* < 0.05).

In terms of the linear bivariate correlation between the concentrations of Zn, Cu, and Fe in the aromatic herbs and the microbial growth of the six groups of foodborne pathogens considered in the study (*L. monocytogenes*, *C. perfringens*, *B. cereus*, psychrophilic microorganisms, mesophilic microorganisms, and molds and yeasts), Spearman’s nonparametric method was used, since a normal distribution of the values of microbial growth analyzed by the Kolmogorov‒Smirnov test (*p* > 0.05) was not observed for any of the six groups of microorganisms analyzed. Table 4 shows the values of the linear correlation coefficients (*r*) and the corresponding significance levels (*p*). It can be seen that the growth rate of *L. monocytogenes*, *C. perfringens*, *B. cereus*, and psychrophilic microorganisms increased, indicating that it was linearly and significantly correlated with the concentrations of Zn, Cu, and Fe in the aromatic herbs (*p* < 0.05). This could be associated with a possible stimulating effect of microbial growth in the four groups of food pathogenic microorganisms indicated, exerted by the antioxidant minerals studied (Zn, Cu, and Fe). Additionally, the growth of mesophilic microorganisms and molds and yeasts also increased significantly with the Cu concentrations measured in the aromatic herbs (*p* < 0.001; Table 4).

## 4. Discussion

As we previously indicated, the present concentrations of Zn, Cu, and Fe depend on the aromatic herb considered. We found higher average values in basil and lower average values in cloves (Figure 1A,B), which led us to establish, specifically for Zn and Cu levels, the following decreasing order of basil > oregano, similar to thyme > rosemary > cloves. For mean Zn and Cu concentrations, similar to what we found in the present study, others [40] also found higher and lower concentrations in basil and cloves, respectively. For Zn, others [21] reported concentrations in the order of thyme > rosemary > oregano. For Cu, the descending order for mean concentrations measured by Potorti et al. [21] was oregano > rosemary > thyme. For Fe, the average concentrations were in the order thyme > basil > oregano > rosemary > cloves. Other researchers [21] also found higher Fe concentrations in thyme than in oregano; however, others [2] reported Fe concentrations in the following descending order: oregano > rosemary > basil.

As shown in Table 1, when comparing the findings obtained in this work with those determined in different studies conducted by other researchers in other places, we see a great variability in the concentration of these essential elements (Zn, Cu, and Fe) in the herbs analyzed. The results show that the concentrations of the minerals studied in the herbs depend on their geographical origin, as the soil and the specific climatic conditions of that area [20] determine the final amount of Zn, Cu, and Fe present.

For Zn (Table 1), in thyme, rosemary, oregano, and basil in all studies, with the exception of one, higher levels were collected than those determined in this work; in the case of cloves, the two studies considered [1,40] collected average concentrations considerably higher than those measured by us. For Cu, the average concentrations determined in the present work are similar to those measured in two of the studies included in Table 1 for thyme and rosemary, and it is remarkable that in basil they are higher than those determined in the four different studies included (Table 1). For Fe, the average concentrations determined in the present work are considerably lower in rosemary, cloves, oregano, and basil than in the studies carried out by other authors (Table 1); in the case of thyme, in only two of the five research works included, the measured concentrations determined were lower than those measured by us. Therefore, globally, the average concentrations of Zn and Fe determined in this study in the aromatic herbs considered are low, lower than those measured by other authors, and met the safety standards, as others [1] stated for cloves, among other spices and herbs.

The levels of the evaluated metals in the considered herbs have been correlated. It has been observed that the concentrations of Zn correlate linearly and significantly with those of Fe (*r* = 0.666, *p* < 0.001) and Cu (*r* = 0.733, *p* < 0.001), as well as those of Fe with Cu (*r* = 0.667, *p* < 0.001). Similarly, in some herbs and herbal teas, others [41] found linear and positive correlations between Zn and Fe and Cu, as well as between Fe and Cu.

According to various researchers [1,24,45], the absorption of potentially toxic elements by plants (such as aromatic herbs), especially due to the presence of high concentrations in the soils of origin, which could occur for Zn, Cu, and Fe, is an important route by which they can enter the food chain, which can lead to bioaccumulation in these herbs and after human consumption, causing toxic effects. In this sense, Sabina et al. [24] indicate that an excess of iron would act as a pro-oxidant and could be toxic to cells, promoting the generation of free radicals that damage their lipid membranes, proteins, and nucleic acids. However, the low average levels of Zn, Cu, and Fe in the analyzed aromatic herbs do not constitute toxic effects of these metals for human beings. On the contrary, as essential elements, as we will refer to them later, their daily intake in the diet does not cover more than 1.6% of the recommended daily dietary intake for healthy adult men and women [22]. Other authors [21,47,48] point out that some heavy metals are present in spices and aromatic plants in various concentrations, due to the use of contaminated irrigation water, the addition of fertilizers and herbicides to the soil to grow crops, and aerial deposition in the plant foliage from environmental pollutants derived from the use of fossil fuels and industrial activities.

Given the essential nature of these elements, it has been estimated that a healthy adult man and woman could get a certain percentage of their respective recommended daily intake, as established by the Institute of Medicine, through their daily diet [22]. Taking into account the average concentrations determined for Zn, Cu, and Fe in all the dehydrated aromatic herbs analyzed, with values of 11.95 ± 0.509, 6.85 ± 0.487, 82.6 ± 5.67 g/g, respectively, and the fact that the estimated average daily intake is very low (1.5 g/day), the daily intake would be 0.018, 0.010, and 0.124 mg/day, which is 0.164%, 1.13%, and 1.55% of the recommended dietary intake for a healthy adult male, and 0.212%, 1.13%, and 0.689% of the recommended dietary intake for a healthy adult female, respectively [22]. Others [21] also reported that the intake of essential elements (like Zn, Cu, and Fe) through spices and aromatic herbs from Sicily (Italy) and Mahdia (Tunisia) was small.

Of all the dehydrated herbs analyzed, cloves are the herb with the best hygienic quality, with no detectable microbial growth in *L. monocytogenes*, *C. perfringens*, and *B. cereus* (Figure 2A,B and Figure 3A, respectively). Other authors pointed out the antimicrobial effect of Zn oxide nanoparticles reinforced with extracts of cloves and cinnamon against oral pathogens [11], so they could be used as an alternative to the commercial antimicrobial preservatives available. Other authors [49] also proved the antimicrobial effect of clove essential oils, in the form of composite films loaded with 50% essential oils in combination with Zn oxide nanorods, both in an in vitro model and in a food model of peeled shellfish packaged during refrigerated storage (20 days), especially against *L. monocytogenes* and *Salmonella thyphimurium* previously inoculated into the food. On the contrary, basil is the herb that presents the highest microbial counts in *C. perfringens*, *B. cereus*, mesophilic microorganisms, and molds and yeasts, surpassing in the case of *B. cereus* the doses considered as infective.

However, there are numerous studies that speak of the powerful antimicrobial effect against Gram-positive bacteria presented by essential oils and clove extracts [50,51]. Eugenol, the major component of clove essential oil, together with carvacrol, acts by inhibiting the production of extracellular enzymes, such as amylases and proteases, which mostly cause cellular lysis of microorganisms such as *L. monocytogenes* [52]. Perhaps this is why, in our study, of all the samples analyzed, the one with the lowest microbial counts was cloves, whether sold in bulk or packaged in PET or glass. This would indicate that this spice could be used in the seasoning of foods that are going to be consumed raw, without giving rise to an eventual foodborne disease, which does not happen with the other herbs studied in this work, where microbial counts in molds and yeasts, *B. cereus*, *L. monocytogenes*, and *C. perfringens* are very high.

In relation to the hygienic quality of the samples analyzed, our study showed that the samples sold in bulk presented a high degree of contamination with respect to *L. monocytogenes*, *B. cereus*, psychrophilic and mesophilic microorganisms, and molds and yeasts, with thyme, rosemary, and basil presenting the highest levels of microbial contamination. These results agree with those obtained by Sospedra et al. [27], who found a high degree of contamination by food pathogens in samples sold in street stalls, possibly due to increased environmental contamination, unsanitary hygienic conditions, and/or incorrect handling at these stalls.

Regarding *L. monocytogenes*, mean microbial growth values were significantly higher in herbs sold in bulk compared to those in glass containers. *Listeria* is another bacterium that indicates a lack of hygienic quality in food establishments, since it is an environmental pathogen that is capable of forming biofilms on the surfaces to which it adheres. This result coincides with that of other authors [53], who point out that the risk of consuming food contaminated with this microorganism is two to three times greater when it is purchased from street vendors.

Among the main factors contributing to the contamination of food by *B. cereus* are contaminated food equipment and facilities, and poor hygiene conditions in food processing and preparation sites [54]. This could be why this microorganism presents high levels of growth in general in herbs sold in bulk, as we have seen in our study (Table 3).

In terms of mesophilic aerobic microorganisms, we found that oregano has the lowest microbial counts, significantly lower than those of thyme, rosemary, and basil. However, other researchers [55] report higher mesophilic microorganism counts in salmon and algae burgers treated with oregano essential oils than with thyme essential oils.

In the growth counts of *C. perfringens* we carried out, basil had the highest microbial growth, followed by thyme and rosemary (Figure 2B). However, for essential oils from the same herbs, other researchers [56] reported that, although all their essential oils are bactericides, the minimum inhibitory concentration against *C. perfringens* is different, being lower for thyme (1.25 mg/mL), followed by basil (5 mg/mL), and higher for rosemary (10 mg/mL). Silva et al. [57] compared the antimicrobial activity of essential oils from 10 aromatic plants against different food-contaminating bacteria such as *L. monocytogenes*, *C. perfringens*, and *B. cereus*, among others, and concluded that the most active essential oils are those of thyme and oregano, while the least active is that of basil, which does not even have an inhibitory effect on the growth of *L. monocytogenes*. It is possible that, in the dried samples used in our work, the very low existing concentration of essential oils is not directly related to the greater or lesser microbial growth, hence the discrepancy observed with the studies referred to above.

On the other hand, basil presents the highest counts of *C. perfringens* (Figure 2B), *B. cereus* (Figure 3A), psychrophiles (Figure 3B), mesophilic microorganisms (Figure 4A), and molds and yeasts (Figure 4B), exceeding in the case of *B. cereus* the doses considered as infective. The antifungal and antibacterial activity of basil has been highlighted by various authors [16,58,59], although all agree that this effect is due to the high linalool content of the essential oil obtained from basil extracts and not to the herb itself. In our study, we can see how the use of this dehydrated herb for the sale in bulk of foods that are not going to undergo any thermal treatment before consumption (as is the case with salads of vegetables and pasta flavored with basil, ready-to-eat foods, and certain raw-cured meat derivatives) could eventually lead to a foodborne disease, especially due to the high microbial load that this aromatic herb presents of *B. cereus* and *C. perfringens* and sporulated microorganisms capable of surviving the processing. This, together with inadequate preservation [60,61], as happens in bulk sale, would explain the high microbial counts found in the samples we analyzed (Table 3). Therefore, these findings show the importance of adequate hygiene practices, and measures to be taken in the harvesting of aromatic herbs, such as basil, until they reach the final consumer, as others previously indicated for traditional herbs consumed in the United Arab Emirates [62].

In this work, we have verified that in the quantities in which they are habitually used for culinary purposes, the very low level of essential oils (data not evaluated; this will be undertaken in future studies) does not exert the expected and reported antimicrobial effect in several research studies [6,7,8,9,10,11,12,15,16,17,32,33,51,52,56,57,58,59]. In these studies, the essential oils extracted from these aromatic herbs are used directly, leading to the proposal that they might be a natural alternative to inorganic chemical preservatives. On the other hand, if the conditions of storage, sale, handling, technological processing, etc., are not adequate, these aromatic herbs can act as vehicles for pathogenic microorganisms [27,54] responsible for foodborne diseases, as we have shown in this study, mainly for basil.

As reported by several authors [14,63], both glass and PET packaging systems have excellent barrier properties against gases such as oxygen and carbon dioxide due to the dense packaging of their molecules, which makes gas permeability difficult. This, together with their barrier properties against microorganisms from the environment [64], could explain why, in our study, the average values of microbial growth were significantly higher in aromatic herbs sold in bulk than in those in glass and PET containers for psychrophilic microorganisms, molds and yeasts, and compared to those sold in glass containers for *L. monocytogenes*, *B. cereus*, and mesophilic microorganisms.

For Miranda et al. [65], the quality of nuts packed with apricots and raisins depends on the free space inside the container and the permeability to water and oxygen through the container during storage. For these authors, glass is an excellent material for protecting the quality of these foods due to its chemical inertness and impermeability. One of the reasons why PET has displaced glass is because of its lightness, which allows for a reduction in transport costs. In our study, the commercialization of the analyzed herbs in glass and PET containers improves the hygienic quality with respect to those sold in bulk. There are no significant statistical differences between the different microorganisms studied according to the packaging system (glass and PET), with the exception of molds and yeasts.

PET is more impermeable to carbon dioxide than to oxygen, which allows the oxygen level in the headspace to increase during the first months of storage until the concentration of external oxygen is reached. This could explain why, in our study, the samples packaged in PET had lower microbial counts than those sold in bulk, in terms of psychrophilic microorganisms and molds and yeasts, since PET can passively create a beneficial modified atmosphere inside the package, establishing a balance between the intensity of breathing of the product and the transmission of oxygen and carbon dioxide through the film, which translates into a lower intensity of microbial growth by slowing it down [66].

In the present study, we have verified that the growth of *L. monocytogenes*, *C. perfringens*, *B. cereus*, and psychrophilic microorganisms increases with the concentrations of Zn, Cu, and Fe in the dehydrated aromatic herbs considered (Table 4). In addition, we verified that basil has the highest concentrations of Zn, Cu, and Fe (Figure 1A,B), and the microbial count of *C. perfringens*, *B. cereus*, psychrophilic microorganisms, mesophilic microorganisms, and molds and yeasts was greatest (Figure 2B, Figure 3A,B and Figure 4A,B, respectively). In the case of *L. monocytogenes* (Figure 2A), its growth was greater in basil than in the other aromatic herbs, with the exception of thyme. For all these reasons, besides the greater possible contamination of basil by microorganisms and its action as a vehicle to the microbial culture medium, we could not ignore the possibility that the greater content of Zn, Cu, and Fe in this aromatic herb could act as a growth factor for the food pathogenic microorganisms analyzed.

The lowest concentrations of Zn, Cu, and Fe were found in cloves (Figure 1A,B), and there was no microbial growth of *L. monocytogenes*, *C. perfringens*, and *B. cereus*. Moreover, in cloves, the development of psychrophilic microorganisms was lower than in thyme and basil, the growth of mesophilic microorganisms was lower than in basil, and the development of molds and yeasts was lower than in rosemary and basil. For all these reasons, in addition to the lower microbial contamination of cloves, we could not rule out the possibility that the lower content of Zn, Cu, and Fe in this aromatic herb could lower the risk of these food pathogenic microorganisms, as the herb does not present adequate levels of Zn, Cu, and Fe to promote growth. To clarify this aspect, future studies should be planned in which increasing amounts of Zn, Cu, and Fe are added to the culture medium, together with the different aromatic herbs, to evaluate to what extent this is related to increased microbial growth. This would allow us to determine the quantity of these minerals to add to the culture medium to exert a growth-stimulating effect.

## 5. Conclusions

The content of Zn, Cu, and Fe varies depending on the aromatic herb. The mean Zn and Cu concentrations were different among all aromatic herbs, with the exception of thyme and oregano. The highest mean Zn and Cu levels were found in basil, higher than thyme and oregano, which were higher than that measured in rosemary, which was higher than that determined in cloves. The average levels of Fe determined in thyme and basil are similar and higher than those of the other aromatic herbs. The highest mean concentrations were found in basil for Zn and Cu, and in thyme and basil for Fe, and the lowest ones in cloves for these three analyzed minerals. The commercialization system (glass or PET packaging, or bulk sale) does not influence the content of Zn and Cu in the analyzed aromatic herbs. The Fe levels determined in herbs marketed in PET were lower than those marketed in glass or in bulk.

After the addition of the aromatic herbs to the microbial culture medium, basil was the herb for which the microbial count was highest in five of the six foodborne pathogens studied, namely *C. perfringens*, *B. cereus*, psychrophilic microorganisms, mesophilic microorganisms, and molds and yeasts. Basil, most of all, is the herb that could eventually lead to a foodborne disease from its sale in bulk, as when added to foods that are not going to suffer any thermal treatment before consumption (salads, pasta, ready-to-eat foods, and certain raw-cured meat derivatives). On the contrary, cloves showed no microbial growth of *L. monocytogenes*, *C. perfringens*, and *B. cereus*, and therefore have the best hygienic quality and could be used as a natural preservative in food. Aromatic herbs marketed in bulk showed a higher microbial count compared to PET and glass packaging for psychrophilic microorganisms, and molds and yeasts, and compared to glass packaging for *L. monocytogenes*, *B. cereus*, and mesophilic microorganisms. The sale of the analyzed herbs in glass and PET containers improves the hygienic quality with respect to those sold in bulk.

The Zn, Cu, and Fe concentrations were positively correlated with microbial growth for *L. monocytogenes*, *C. perfringens*, *B. cereus*, and psychrophilic microorganisms (*p* < 0.05) so they could act as a growth factor for foodborne pathogens.

## Figures and Tables

**Figure 1 foods-09-01658-f001:**
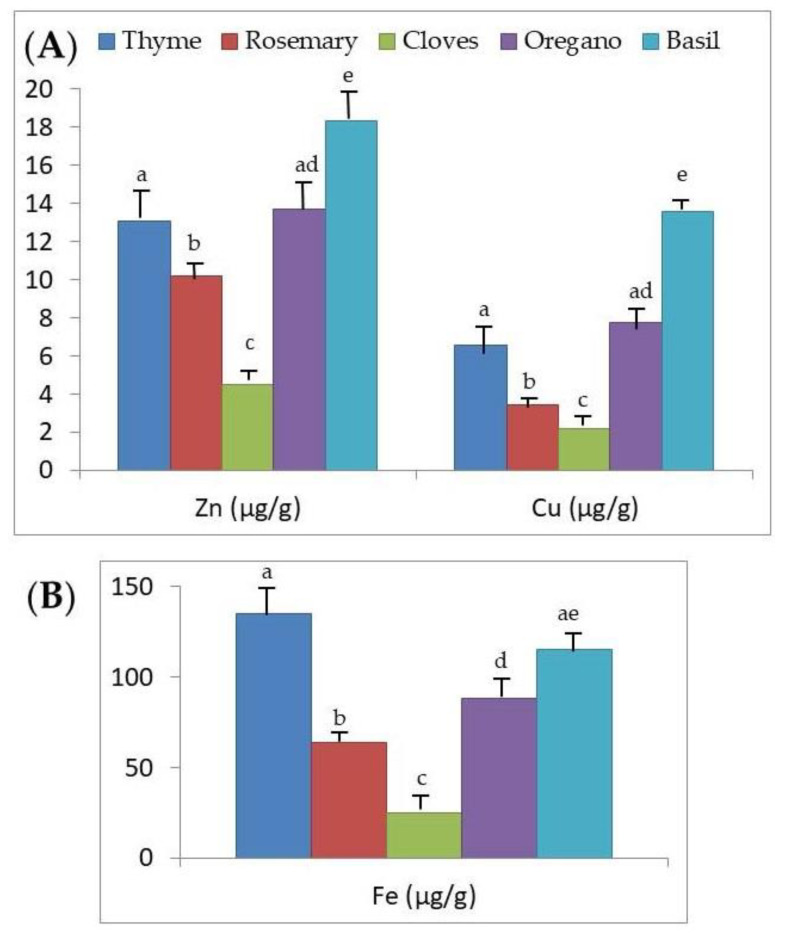
Mean (**A**) Zn, Cu, and (**B**) Fe concentrations (μg/g) measured in dehydrated aromatic herbs (thyme, rosemary, cloves, oregano, and basil). Mean Zn, Cu, and Fe values; different superscripts indicate significant differences between the aromatic herbs (*p* < 0.001).

**Figure 2 foods-09-01658-f002:**
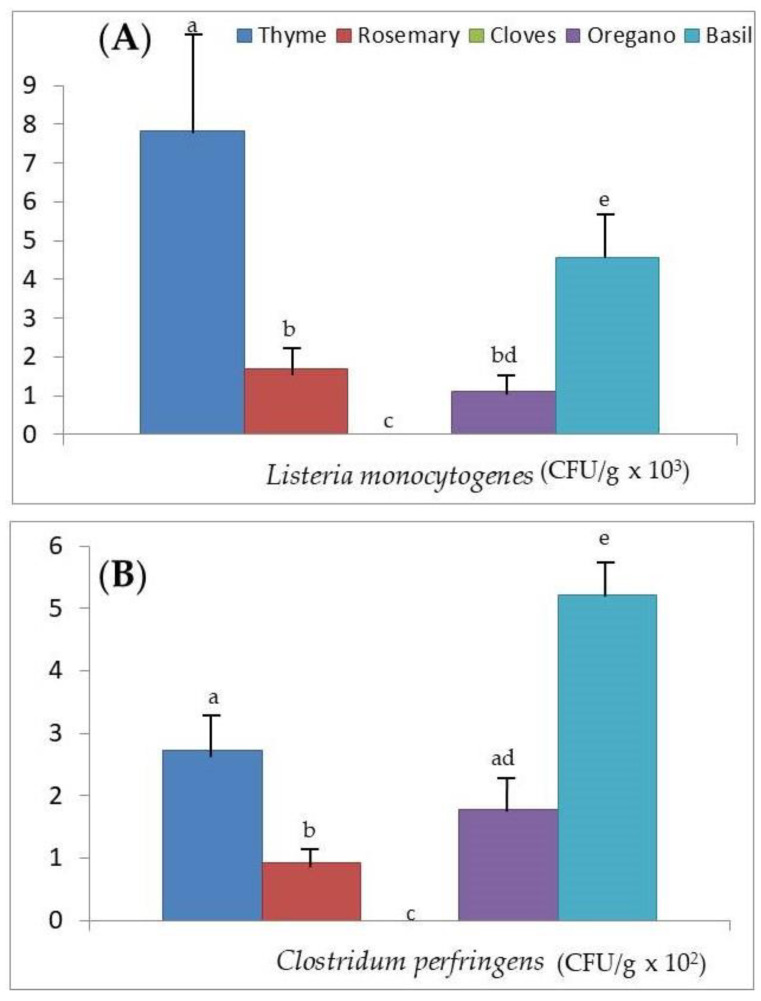
Microbial counts measured as colony forming units (CFU/g) for (**A**) *Listeria monocytognes* and (**B**) *Clostridium perfringens* after the addition of the dehydrated aromatic herbs (thyme, rosemary, cloves, oregano, and basil) to the culture medium. ^a,b,c,d,e^ Mean microbial counts; different superscripts indicate significant differences between the aromatic herbs (*p* < 0.01).

**Figure 3 foods-09-01658-f003:**
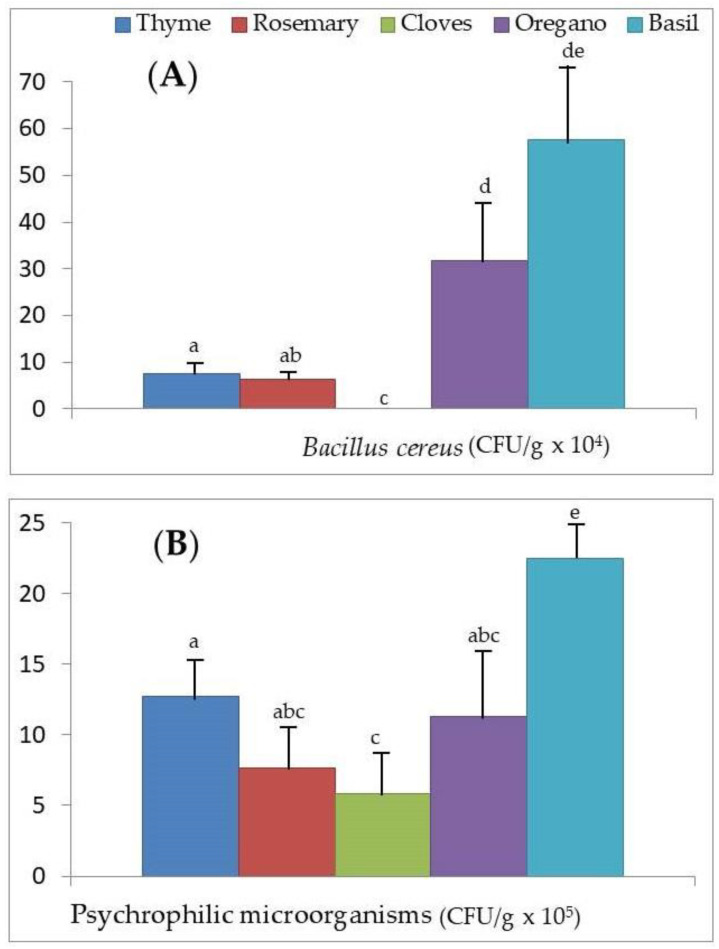
Microbial counts measured as colony-forming units (CFU/g) for (**A**) *Bacillus cereus* and (**B**) psychrophilic microorganisms after the addition of the dehydrated aromatic herbs (thyme, rosemary, cloves, oregano, and basil) to the culture medium. ^a,b,c,d,e^ Mean microbial counts; different superscripts indicate significant differences between the aromatic herbs (*p* < 0.001).

**Figure 4 foods-09-01658-f004:**
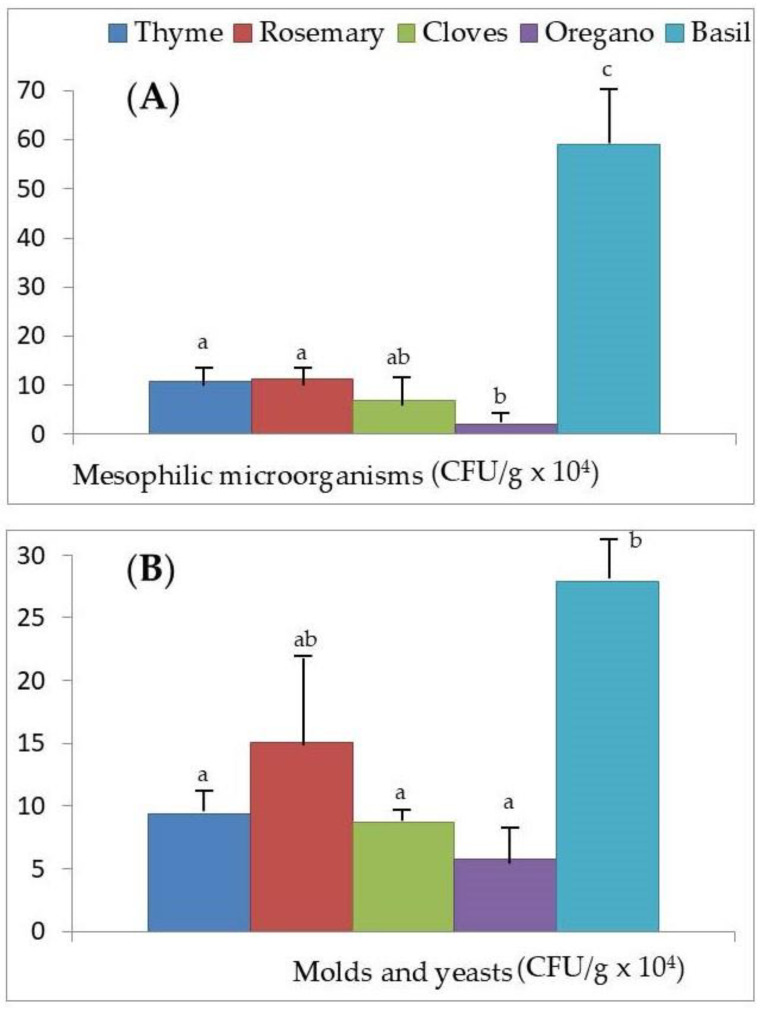
Microbial counts measured as colony forming units (CFU/g) for (**A**) mesophilic microorganisms and (**B**) molds and yeasts after the addition of the dehydrated aromatic herbs (thyme, rosemary, cloves, oregano, and basil) to the culture medium. ^a,b,c,d,e^ Mean microbial counts; different superscripts indicate significant differences between the aromatic herbs (*p* < 0.001).

**Table 1 foods-09-01658-t001:** Mean Zn, Cu, and Fe concentrations (μg/g) and standard error of the mean (SEM) in commercial samples of dehydrated herbs (thyme, rosemary, cloves, oregano, and basil) measured in the present study in comparison to those determined by other researchers.

Aromatic Herb	Zn ± SEM	Cu ± SEM	Fe ± SEM	Reference
Thyme	-	-	857 ± 57.9	[2]
Thyme	330 ± 67	26 ± 4.1	41 ± 11	[21]
Thyme	17 ± 1.7	1.9 ± 0.64	112 ± 13	[40]
Thyme	22 ± 2.3	6.1 ± 1.9	440 ± 1.4	[41]
Thyme	5.4 ± 1.6	6.6 ± 0.32	203 ± 1.2	[42]
Thyme	13 ± 0.75	6.6 ± 0.75	135 ± 11	Present study
Rosemary	-	-	118 ± 16.6	[2]
Rosemary	41 ± 7.6	84 ± 14	n.d.^a^	[21]
Rosemary	31 ± 3.2	3.0 ± 0.02	735 ± 39	[40]
Rosemary	9.0 ± 0.30	5.0 ± 1.0	173 ± 3.2	[43]
Rosemary	52	8.5	432	[44]
Rosemary	10 ± 0.23	3.4 ± 0.16	63 ± 7.6	Present study
Cloves	6.3 ± 0.90	3.2 ± 0.10	90 ± 6.0	[1]
Cloves	14 ± 1.7	1.1 ± 0.02	65 ± 3.6	[40]
Cloves	4.5 ± 0.19 ^c^	2.2 ± 0.11 ^c^	25 ± 5.9 ^c^	Present study
Oregano	-	-	918 ± 44	[2]
Oregano	31 ± 8.5	85 ± 2.7	n.d.	[21]
Oregano	59 ± 6.8	8.4 ± 6.5	240 ± 217	[24]
Oregano	9.0 ± 0.40	3.0 ± 1.0	198 ± 2.4	[43]
Oregano	14 ± 0.46	7.7 ± 0.36	88 ± 6.4	Present study
Basil	-	-	112 ± 65.1	[2]
Basil	43 ± 1.4	7.6 ± 0.06	251 ± 38	[40]
Basil	46 ± 1.5	4.8 ± 0.15	448 ± 12	[42]
Basil	16 ± 0.20	11 ± 0.30	390 ± 14	[43]
Basil	35 ± 4.1	6.8 ± 1.5	671 ± 20	[45]
Basil	89	138	305	[46]
Basil	18 ± 0.48	14 ± 0.42	115 ± 7.6	Present study

^a^ Not detected.

**Table 2 foods-09-01658-t002:** Mean Zn, Cu, and Fe concentrations (μg/g) and standard error of the mean (SEM) in herbs (thyme, rosemary, cloves, oregano, and basil) according to the marketing system (in bulk, or in packaging in glass or polyethylene terephthalate (PET)).

Marketing System	Zn ± SEM	Cu ± SEM	Fe ± SEM
PET	12 ± 0.95 ^a^	5.6 ± 0.72 ^a^	53 ± 9.8 ^a^
Glass	12 ± 1.0 ^a^	5.5 ± 0.70 ^a^	90 ± 8.0 ^b^
Bulk	12 ± 0.72 ^a^	8.5 ± 0.69 ^a^	90 ± 8.7 ^b^

^a,b,c^ Mean Zn, Cu, and Fe concentrations with different superscripts express significant differences between different marketing systems (*p* < 0.01).

**Table 3 foods-09-01658-t003:** Microbial count values (CFU/g) and standard error of the mean (SEM) for *Listeria monocytogenes*, *Clostridium perfringens*, *Bacillus cereus*, psychrophilic microorganisms, mesophilic microorganisms, and molds and yeasts in culture media previously infused with herbs (thyme, rosemary, cloves, oregano, and basil) according to the marketing system (in bulk, or in packaging in glass or PET).

Microorganisms (CFU/g)	Packaging in Glass (Mean ± SEM)	Packaging in PET (Mean ± SEM)	In Bulk (Mean ± SEM)
*L. monocytogenes*	71 ± 17 ^a^	601 ± 223 ^ab^	5119 ± 921 ^b^
*C. perfringens*	142 ± 24 ^a^	172 ± 36 ^a^	274 ± 38 ^a^
*B. cereus*	198,936 ± 77,352 ^a^	55,286 ± 18,921 ^ab^	286,851 ± 79,455 ^b^
Psychrophilic microorganisms	287,361 ± 62,082 ^a^	1,245,643 ± 266,560 ^a^	2,488,507 ± 382,092 ^b^
Mesophilic microorganisms	14,403 ± 4537 ^a^	324,500 ± 104,357 ^ab^	184,792 ± 25,579 ^b^
Molds and yeasts	73,833 ± 14,560 ^a^	32,143 ± 13,416 ^b^	209,306 ± 24,368 ^c^

^a,b,c^ Mean microbial counts values for studied microorganisms; different superscripts indicate significant differences between different marketing systems. ^a,b,c^ Mean microbial counts values were significantly higher in aromatic herbs sold in bulk vs. those packaged in glass and PET for psychrophilic microorganisms, and molds and yeasts and vs. those in glass for *L. monocytogenes*, *B. cereus*, and mesophilic microorganisms (*p* < 0.05).

**Table 4 foods-09-01658-t004:** Linear correlation coefficients (*r*) and significance levels (*p*) between measured Zn, Cu, and Fe concentrations in the herbs and the microbial count values (CFU/g) for *Listeria monocytogenes*, *Clostridium perfringens*, *Bacillus cereus*, psychrophilic microorganisms, mesophilic microorganisms, and molds and yeasts in culture media previously infused with those herbs.

Microorganisms	Zn		Cu		Fe	
	*r*	*p*	*r*	*p*	*r*	*p*
*L. monocytogenes*	0.264	0.012	0.559	0.001	0.427	0.001
*C. perfringens*	0.632	0.001	0.775	0.001	0.655	0.001
*B. cereus*	0.253	0.017	0.428	0.001	0.356	0.001
Psychrophilic microorganisms	0.267	0.011	0.546	0.001	0.375	0.001
Mesophilic microorganisms	0.204	0.055	0.383	0.001	0.097	0.377
Molds and yeasts	0.125	0.242	0.435	0.001	0.201	0.065

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
