# Peer review of "Zn, Cu, and Fe Concentrations in Dehydrated Herbs (Thyme, Rosemary, Cloves, Oregano, and Basil) and the Correlation with the Microbial Counts of Listeria monocytogenes and Other Foodborne Pathogens"

_foods, 2020, doi:10.3390/foods9111658_

Round 1

Reviewer 1 Report

The paper describes the correlation between the Zn, Cu and Fe concentrations and the presence of Food Borne Pathogens, in dehydrated herbs.

Despite the paper is adequately argued and includes good references, it is necessary to clarify some aspects in the Materials and methods paragraph.

The authors focus their evaluation on mesophilic, psychrophilic, B.cerus, Cl.perfringens and L. monocytogenes microorganisms.

Psychrophilic bacteria  are defined as "cold-loving bacteria".  The optimal temperature for maximal growth is 20 °C, why did you use a incubation temperature of 30°C?

For yeasts and moulds the Authors refer to apply ISO 21527-2: 2008, but the culture medium indicate by this procedure is only DG 18; Why did you use Sabouraud Dextrose Agar?   The inoculation of plates by pour-plate methods shall be validated compared to inoculation on the surface, have, the Authors, performed this activity?

For Listeria monocytogenes enumeration the authors apply ISO 11290-2, but the method described not correspond with what is indicated by the cited standard. The starting dilution is excessive, the incubation times too long for a count in microbiology (max 45 'from ISO 11290-2).

Author Response

First of all we would like to thank reviewers by their comments and suggestions because we consider that they really increase the scientific value of this study, as well as help to clarify some aspects of the manuscript. All changes done in the revised version of the manuscript have been highlighted in yellow in order to facilitate reviewers and editorial board their location along the text. In relation to this, we would like to make the following considerations:

First of all, following reviewer’s comments we have asked for the revision of the whole English edition of the manuscript to the MDPI English editing service, after uploading the revised manuscript to the webpage of Foods.

-    Psychrophilic bacteria are defined as "cold-loving bacteria".  The optimal temperature for maximal growth is 20 °C, why did you use a incubation temperature of 30°C?

     *In fact, psychophilic microorganisms have their maximum growth rate around 13ºC. In our study, after sowing the samples in plates with KING A Agar, they were left at room temperature (20ºC, 24 h). After, the colonies in the culture plates were counted. This was a transcription error in the manuscript. Thank you for warning us of this mistake. Therefore in the revised version we have properly changed to “Agar medium is used followed by incubation for 24-48 hours at room temperature (20°C)” (lines 188-189).

-    For yeasts and moulds the Authors refer to apply ISO 21527-2: 2008, but the culture medium indicate by this procedure is only DG 18; Why did you use Sabouraud Dextrose Agar? The inoculation of plates by pour-plate methods shall be validated compared to inoculation on the surface, have, the Authors, performed this activity?

     * For the counting of molds and yeasts we have followed the technique described in the ISO 21527: 2008 Standard, but we have replaced the Dicloren Glycerol DG 18 Agar culture medium by the Sabouraud Dextrose Agar since we had enough amount of this medium in the laboratory and we always use it to determine molds and yeasts according to the Standards learned in the National Center of Microbiology (Carlos III Institute, Madrid, Spain) for food and beverages. Consequently we have changes the revised version to “Yeast and mold enumeration was performed by the pour-in-plate method using Sabouraud agar with chloramphenicol (Sabouraud Dextrose Agar, according to the standards learned for food and beverages in the National Center of Microbiology, Carlos III Institute, Madrid, Spain), followed…” (lines 186-187).

     *The inoculation of plates by pour-plate methods was not validated compared to inoculation on the surface, because the used Sabouraud Dextrose Agar is specific of pour-plate method.

-    For Listeria monocytogenes enumeration the authors apply ISO 11290-2, but the method described not correspond with what is indicated by the cited standard. The starting dilution is excessive, the incubation times too long for a count in microbiology (max 45 'from ISO 11290-2).

     *Following 2nd reviewer’s comments this paragraph has been chand in the revisd version of the manuscript toIn order to detect L. monocytogenes, according to ISO 11290:2017 [37], the studied samples (1.5 g) were weighed in Stomacher bags, diluted and homogenized with 225 mL of Fraser (primary enrichment in Half-Fraser broth: incubation for 25 ± 1 h); secondary enrichment in Fraser broth: incubation for 24 ± 2 h. After this time, sowing was carried out in Listeria plates of Palcam Listeria Agar 37 ± 1ËšC during 24 ± 2 h (plus another 24 ± 2 h at 37 ± 1ËšC ).” (lines 190-194).

     *In relation with the comment done by 2nd reviewer about that the starting dilution is excessive, we have to say that our own experience have demonstrated us that this dilution is the appropriate in our experimental conditions in this type of samples.

Reviewer 2 Report

In the submitted work the Authors analyze the concentrations of three selected metals in the studied dehydrated herbs and try to correlate them with the microbiological results. The submitted study is rather routine and the results could have been easily predicted, based on the common knowledge. My comments are listed below.

Line 18: This sentence should be removed. It is too general for an abstract.

Line 21: “Microbial counts of Listeria monocytogenes and other five food-borne pathogens when herbs were previously added to the growing media were also checked, to evaluate them as vehicles of these microorganisms to food.” This sentence makes no sense. It is grammatically incorrect.

Keywords should be redefined.

Line 37: Not “Its” but “Their”.

Line 63: It should be “nano Zn”.

Line 67: What do you mean by “inorganic elements”

Line 71: Not “minerals” but “metals”.

Line 73: Why “Institute of Medicine” is chosen as an reference-giving organization?

Lines 77-83: This part is too long.

Lines 115-116: It is known for a long time that those herbs have the antimicrobial effects, there is nothing more to prove in this topic.

Line 170: “The microorganisms whose microbial growth has been evaluated have been” – this is incorrect. There are also other grammar issues like this one.

Statistical analysis: the Authors should have used ANOVA.

Table 2: SEM should be explained.

Figure 1: Please use colors.

Table 3 but also other Tables. The numbers are not rounded correctly.

Line 366: Not “minerals” but “metals”.

Line 387: Why “Institute of Medicine” is chosen as an reference-giving organization?

Line 409: “major”, not “majority”.

Line 523: Remove one dot.

Lines 555-565: This should be removed.

Author Response

First of all we would like to thank reviewers by their comments and suggestions because we consider that they really increase the scientific value of this study, as well as help to clarify some aspects of the manuscript. All changes done in the revised version of the manuscript have been highlighted in yellow in order to facilitate reviewers and editorial board their location along the text. In relation to this, we would like to make the following considerations:

First of all, following reviewer’s comments we have asked for the revision of the whole English edition of the manuscript to the MDPI English editing service, after uploading the revised manuscript to the webpage of Foods.

- General comment

* Unfortunately we do not agree with comments of the 2nd reviewer in relation that “this study is rather routine and the results could have been easily predicted, based on the common knowledge”. After the reviewing the existing bibliography in this subject no studies trying to correlate metal levels (Zn, Cu and Fe) in dehydrated aromatic herbs and microbial growth have been found. Most of them have been focused in the metal concentrations in aromatic herbs, and another high number of them in the antimicrobial activity of essential oils isolated from different spices and herbs.

- Line 18: This sentence should be removed. It is too general for an abstract.

* The referred sentence “The herbs have been used by their organoleptic and antimicrobial characteristics in foods” has been erased in the revised version following reviewer’s comments (Line 18).

-    Line 21: “Microbial counts of Listeria monocytogenes and other five food-borne pathogens when herbs were previously added to the growing media were also checked, to evaluate them as vehicles of these microorganisms to food.” This sentence makes no sense. It is grammatically incorrect.

* This sentence has been corrected in the revised version to “Microbial counts of Listeria monocytogenes and other five food-borne pathogens when herbs were previously added to the growing media were also checked” (lines 20-21).         

-    Keywords should be redefined.

* Following reviewer’s comments keywords have been redefined to “Zn, Cu and Fe   concentrations; dehydrated herbs; microbial counts for food-borne pathogens” (lines 30-31).

-    Line 37: Not “Its” but “Their”.

* The word has been properly corrected in the revised version to “Their” (line 35). 

-    Line 63: It should be “nano Zn”.

* The word has been properly corrected in the revised version to “nano Zn” (line 61).

  • Line 67: What do you mean by “inorganic elements”

* We really mean to “elements”, as it has been corrected in the revised version of the manuscript (line 66).

-    Line 71: Not “minerals” but “metals”.

     * The word has been properly corrected in the revised version to “metals” (line 69).

  • Line 73: Why “Institute of Medicine” is chosen as an reference-giving organization?

* We have chosen the Institute of Medicine, because is a reference organization recognized along the world.

-    Lines 77-83: This part is too long.

     * Following 2nd reviewer’s comments the referred part has been shortened to “Zn and Cu, and Fe, are co-factors of the superoxide dismutase, and catalase antioxidant enzymes, respectively. However, high levels of free Cu and Fe in the body act as pro-oxidants [25] resulting in pathologies like cardiovascular diseases, hypertension, type 2 diabetes mellitus, multiple sclerosis, etc. [24]” (lines 76-79, revised version of the manuscript).

  • Lines 115-116: It is known for a long time that those herbs have the antimicrobial effects, there is nothing more to prove in this topic.

* We agree with the reviewer’s statement on that the antimicrobial effects of these herbs are known for a long time. However, most of studies have been performed taking into account the essential oils present in them which are the main responsible of this preservative effect. Nevertheless, the comparison of the preservative capacity of these studied in this manuscript when added to food not submitted to thermal final treatment, has not been profusely performed. Contrarily, even the addition of some of these herbs can act as vehicle of food-borne pathogens, such as we have demonstrated in the present study overall for basil. Taking into account reviewer’s statements we have changed these sentences in the revised version to “As it is known, aromatic herbs have traditionally been associated with and antimicrobial effect overall by their content in essential oils. Nevertheless, the comparison of the preservative capacity of those studied in this manuscript when added to food not submitted to thermal final treatment, has not been profusely performed. Additionally, it has neither been checked if some of them act directly as a vehicle for the arrival of anyone of the groups of these food-borne pathogens” (lines 111-116).

-   Line 170: “The microorganisms whose microbial groof wth has been evaluated have been” – this is incorrect. There are also other grammar issues like this one.

     * This sentence has been corrected in the revised version to “Microbial counts for L. monocytogenes, C. perfringens and B. cereus, as microorganisms indicating lack of hygienic-sanitary quality in herbs, have been checked” (lines 168-169).

-    Statistical analysis: the Authors should have used ANOVA.

     * Evidently when we performed the statistical analysis of data we used the “ANOVA test” for parametric data as it has been corrected in the revised version (line 209).

-    Table 2: SEM should be explained.

     * SEM refers to “standard error of the mean” as it has been explained in tables 1, 2 and 3.

-  Figure 1: Please use colors.

     * Colors have been used in the revised figures as you can check in the revised version.

- Table 3 but also other Tables. The numbers are not rounded correctly.

     * Following 2nd reviewer’s comments the numbers have been correctly rounded in tables 1, 2, and 3, in the revised version of the manuscript.

- Line 366: Not “minerals” but “metals”.

     * It has been changed to “metals” in line 368.

- Line 387: Why “Institute of Medicine” is chosen as an reference-giving organization?

     * Such as we previously reported the Institute of Medicine is a very well recognized international reference organization which establishes the dietary reference intakes for micronutrients for the north-American population. In many of the studies performed by our research group this is the international reference organization used.

-  Line 409: “major”, not “majority”.

     * As requested 2nd reviewer, the word “majority” has been correctly changed to “major” (line 411).

-  Line 523: Remove one dot.

     * One “dot” has been removed in the revised version of the manuscript (line 525).

- Lines 555-565: This should be removed.

     * Requested lines have been erased in the revised version of the manuscript.

Round 2

Reviewer 1 Report

I appreciate the revision of paper and the author's reply

to the referees' comments.  The manuscript has been significantly

improved.

Reviewer 2 Report

The Authors have corrected the manuscript following my instructions.